# Cryo-EM structure of the agonist-bound Hsp90-XAP2-AHR cytosolic complex

Jakub Gruszczyk ®[1] ✉, Loïc Grandvuillemin[1], Josephine Lai-Kee-Him[1], Matteo Paloni ®[1], Christos G. Savva[2], Pierre Germain[1], Marina Grimaldi[3], Abdelhay Boulahtouf[3], Hok-Sau Kwong[1], Julien Bous ®[4], Aurélie Ancelin[1], Cherine Bechara ®[5,6], Alessandro Barducci[1], Patrick Balaguer ®[3] & William Bourguet ®[1] ✉

The aryl hydrocarbon receptor (AHR) is a ligand-dependent transcription factor that mediates a broad spectrum of (patho)physiological processes in response to numerous substances including pollutants, natural products and metabolites. However, the scarcity of structural data precludes understanding of how AHR is activated by such diverse compounds. Our 2.85 Å structure of the human indirubin-bound AHR complex with the chaperone Hsp90 and the co-chaperone XAP2, reported herein, reveals a closed conformation Hsp90 dimer with AHR threaded through its lumen and XAP2 serving as a brace. Importantly, we disclose the long-awaited structure of the AHR PAS-B domain revealing a unique organisation of the ligand-binding pocket and the structural determinants of ligand-binding specificity and promiscuity of the receptor. By providing structural details of the molecular initiating event leading to AHR activation, our study rationalises almost forty years of biochemical data and provides a framework for future mechanistic studies and structure-guided drug design.

Living organisms have developed protein sensors helping them to adapt to their environment[1]. Also known as the dioxin receptor, the aryl hydrocarbon receptor (AHR) is an emblematic member of this class of proteins[2]. AHR is a ligand-dependent transcription factor involved in the regulation of genes governing many (patho)physiological processes including cell growth and differentiation, cell cycle and migration, apoptosis, haematopoiesis and carcinogenesis[3–6]. Importantly, activation of AHR in the gastrointestinal tract participates in signalling between the enteric microflora and our immune system[7–9]. A disruption of the intestinal homoeostasis, for example, has been shown to contribute to the aetiology of many disorders, including inflammatory bowel disease, Crohn's disease, ulcerative colitis, diabetes and obesity[9]. Moreover, AHR is one of the crucial chemosensory proteins associated with the metabolism of various xenobiotics, and sustained activation of the receptor can lead to toxic outcomes[10,11]. AHR belongs to the basic helix-loop-helix (bHLH) PER-ARNT-SIM (PAS) protein family[12]. The bHLH-PAS family, present ubiquitously in all kingdoms of life, is predominantly engaged in detection of such diverse signals like endogenous compounds, foreign chemicals, light and osmotic pressure, for example[2,13]. The numerous cognate ligands of AHR exhibit distinctive chemical structures and include various compounds like polycyclic aromatic hydrocarbons (PAHs), halogenated aromatic hydrocarbons (HAHs), polychlorinated biphenyls (PCBs), indole derivatives, alkaloids, polyphenols and various pharmaceuticals, with dioxin (2,3,7,8-tetrachlorodibenzo-p-dioxin or TCDD) being the archetypal exogenous ligand of AHR[10]. As such, AHR

[1]CBS (Centre de Biologie Structurale), Univ Montpellier, CNRS, Inserm, Montpellier, France. [2]Leicester Institute of Structural & Chemical Biology and Department of Molecular & Cell Biology, University of Leicester, Leicester, UK. [3]IRCM (Institut de Recherche en Cancérologie de Montpellier), Inserm U1194, Univ Montpellier, ICM, Montpellier, France. [4]Section of Receptor Biology & Signaling, Department of Physiology & Pharmacology, Karolinska Institutet, Stockholm, Sweden. [5]IGF, University of Montpellier, CNRS, Inserm, Montpellier, France. [6]Institut Universitaire de France (IUF), Paris, France. ✉e-mail: jakub.gruszczyk@cbs.cnrs.fr; william.bourguet@cbs.cnrs.fr

is a key sensor which integrates numerous external and endogenous chemical signals, and it is now considered a promising drug target[5].

In the absence of ligand, AHR resides in the cytoplasm forming a complex with several other partners including heat shock protein 90 (Hsp90) and co-chaperones like X-associated protein 2 (XAP2, also known as AIP or ARA9) and p23. Upon ligand binding, AHR undergoes conformational changes leading to exposure of the nuclear localization signal (NLS) and therefore triggering a translocation of the complex into the nucleus where AHR is released and interacts with the AHR nuclear translocator (ARNT). The newly formed heterodimer binds to the so-called "dioxin-response element" (DRE, also known as XRE or AHRE) DNA sequences and regulates the expression of target genes[14,15]. Previous structural work revealed how AHR and ARNT heterodimerize and bind to DREs but provided no insights into the AHR cytosolic complex organisation or the structural basis of ligand-binding[16,17]. Here we report the high-resolution cryo-electron microscopy (cryo-EM) structure of the human ligand-bound AHR cytosolic complex as a snapshot of the first step of AHR activation.

## Results

### Structure determination and architecture of the Hsp90-XAP2-AHR complex

Using the Sf9 insect cell expression system, we co-expressed a fragment of human AHR (residues 1-437) in the presence of Hsp90, XAP2 and p23, and purified the Hsp90-XAP2-AHR complex (Supplementary Fig. 1). The initial characterisation of the quaternary Hsp90-XAP2-AHR-p23 complex using cryo-EM indicated a high heterogeneity of the sample due to the presence of several complexes with different number and location of p23 molecules. Therefore, in order to facilitate image processing and improve the resolution of the structure, we decided to use the ternary complex Hsp90-XAP2-AHR. We combined the protein with indirubin and using cryo-EM we obtained a 2.85 Å reconstruction of the ligand-bound AHR cytosolic complex (Fig. 1a, Supplementary Fig. 2, 3 and Supplementary Table 1). The structure reveals in its core a nucleotide-bound Hsp90 dimer (molecules denoted Hsp90A and Hsp90B, respectively) adopting a closed conformation (Fig. 1b and Supplementary Movie 1). Hsp90 forms extensive interaction sites with the PAS-B domain of AHR and XAP2 that occupy the same side of Hsp90, reminiscent of the recently reported structure of the glucocorticoid receptor (GR) complex (Hsp90-p23-GR)[18]. The structure of the bHLH and PAS-A domains of AHR could not be unambiguously resolved during image processing, indicating high dynamics of this region.

The PAS-B domain of AHR exhibits a canonical fold comprising a five-stranded antiparallel β-sheet (Aβ, Bβ, Gβ, Hβ and Iβ) flanked by four consecutive α-helices of variable lengths (Cα, Dα, Eα and Fα). Two

additional short α-helices (Jα and Kα) are present at the carboxy-terminus of the domain (Fig. 2a and Supplementary Fig. 4a). The PAS-B domain is surrounded by additional extensions at both ends. The amino-terminus includes an elongated 15-residue linker region interconnecting PAS-A and PAS-B domains threaded through the Hsp90 dimer lumen (Fig. 2a and Supplementary Fig. 4b, c), and the 40 amino acid residues connecting PAS-B to the carboxy-terminal transactivation domain are forming a long loop that folds back onto the PAS-B domain and participates in the interface with XAP2 (Fig. 2a, b, Supplementary Fig. 4b, c and Supplementary Movie 2). XAP2 interacts with both Hsp90 and AHR (Fig. 1). The amino-terminal domain of XAP2 is flexible and not well resolved in the general map. However, a focused refinement of the cryo-EM data yielded a 4.07 Å resolution map with an improved quality for the whole XAP2 protein and the carboxy-terminus of AHR (Supplementary Fig. 2, 3).

### Hsp90 forms a core of the complex

All components of the complex are forming multiple interaction sites with each other (Fig. 3a). Interaction between AHR and Hsp90 involves two interfaces. The first interface encompasses 731 Å² of buried surface area and includes AHR residues from Aβ (I286-K290), Bβ (G299-D301), Gβ (M348), Hβ (Q364-N366) and Iβ (T382-R384) and Hsp90 residues W312, R338, A339, P340 and F341 from the middle domain of Hsp90A and Y596, T616, Y619 and M620 from the carboxy-terminal domain of Hsp90B (Fig. 3b). Note that residues localised in the β-strand Aβ were previously identified to be important for binding to Hsp90[19].

The second interface, confining 1223 Å² of buried surface area, encompasses the 15-residue linker between the PAS-A and PAS-B domains (P271-F285) that adopts an extended conformation (Fig. 3c and Supplementary Fig. 5a). The linker is threaded through the Hsp90 lumen formed by two Hsp90 molecules, thereby positioning the two PAS domains on the opposite sides of the Hsp90 dimer (Supplementary Fig. 4b). The middle and carboxy-terminal domains from both Hsp90 chains contribute to this interface by providing a combination of hydrophobic and hydrophilic residues that are engaged in multiple contacts with the backbone and side chains of AHR (Fig. 3c and Supplementary Fig. 5a). We observed that AHR I277 and I280 occupy two small hydrophobic pockets within the Hsp90 lumen, similar to L525 and L528 in the Hsp90-p23-GR[18] and to V89 and V92 in the Hsp90-Cdc37-Cdk4[20] structures (Supplementary Fig. 5b, c). Hence, the aliphatic nature of residues at these positions might be a hallmark for client protein interaction with Hsp90. To validate these interactions, we mutated AHR residues P275 to T282 into alanine, and monitored the impact of each mutation on AHR-Hsp90 interaction using co-immunoprecipitation (Co-IP) assays (Fig. 3d and Supplementary Fig. 6a). To this end, we co-transfected HEK cells with plasmids

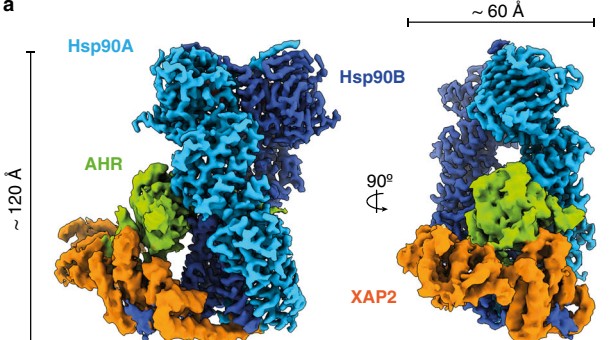

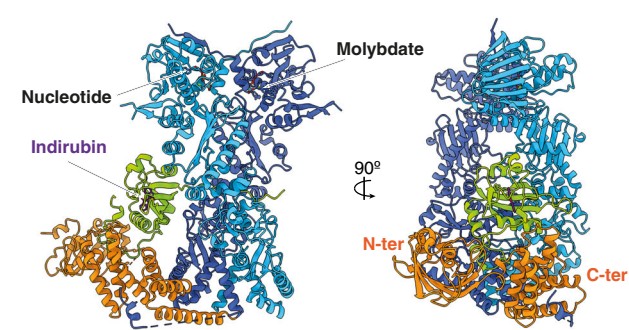

**Fig. 1 | Overall architecture of the agonist-bound cytosolic complex of AHR. a** A composite map of the Hsp90-XAP2-AHR complex in two orthogonal views. Hsp90A (light blue), Hsp90B (dark blue), XAP2 (orange), AHR (green). The same colour scheme is used throughout the manuscript unless stated otherwise. **b** The atomic model of the complex in cartoon representation (the orientation of the molecule same as in a). Nucleotide molecule, molybdate ion and indirubin ligand are shown in sticks.

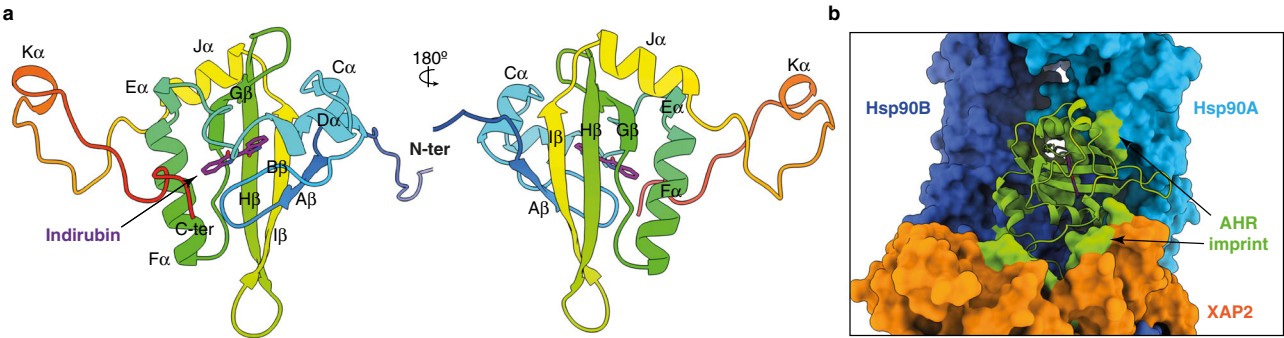

**Fig. 2 | Structure of the AHR PAS-B domain. a** Two opposing views of the PAS-B domain in cartoon representation coloured in rainbow from N- to C-terminus. The elements of the secondary structure are labelled. The indirubin molecule is shown as sticks and coloured in magenta. **b** Close-up view showing PAS-B domain and its interaction with Hsp90 and XAP2. The contact sites of the PAS-B domain with the other components of the complex are coloured in green.

expressing AHR fused with MBP- and Strep-tag, Hsp90 fused with FLAG-tag and XAP2 fused with MYC-tag. The complex was then pulled-down using magnetic beads conjugated with anti-MBP antibody. The presence of the particular components of the complex was detected using the corresponding antibodies. The results revealed that mutation of single residues had a mild effect on the interaction of AHR with Hsp90. Nevertheless, we observed that the most destabilising mutations were those involving the two conserved aliphatic residues (I277, I280) and the polar/charged residues S276 and R281. Intriguingly, these AHR mutations targeting the AHR-Hsp90 interface appeared to have a notable impact on the interaction with XAP2, suggesting a robust interaction of AHR with Hsp90 and somehow weaker association with the co-chaperone.

**Sodium molybdate mimics ATP γ-phosphate**
Sodium molybdate is known to stabilise the client-Hsp90 interaction and to inhibit gene regulation by several nuclear receptors, including AHR[21], but its precise mechanism of action through binding to an ATP site on Hsp90 has remained unproven. Our map displays an unambiguous density for the nucleotide binding site in both Hsp90 chains (Fig. 4a and Supplementary Fig. 7a). We observed an unusually strong density at the expected position of ATP γ-phosphate, much stronger than either α- or β-phosphate (Fig. 4b and Supplementary Fig. 7b). We reasoned that, rather than the ATP γ-phosphate, this density could be that of a molybdate ion that was used in the buffer for complex preparation. We confirmed the presence of the molybdate inside the Hsp90 complex using mass spectrometry (Supplementary Fig. 7d). Therefore, we concluded that the nucleotide binding site contains one ADP molecule generated from ATP hydrolysis, one magnesium ion, and one molybdate ion that is also involved in the chelation of the magnesium ion (Fig. 4c and Supplementary Fig. 7c). Hence, by mimicking the γ-phosphate upon ATP hydrolysis, the molybdate ion would hold the Hsp90 dimer in the ATP-induced closed state, thereby preventing AHR release[22] (Fig. 4d).

**Analysis of the XAP2-AHR and XAP2-Hsp90 interfaces**
The XAP2 protein is composed of an amino-terminal FKBP-type peptidyl-prolyl cis/trans isomerase (PPIase) domain and a carboxy-terminal tetratricopeptide repeat (TPR) helical domain[23,24] (Fig. 5a). Our structure reveals the molecular basis of XAP2 interaction with AHR and Hsp90 suggesting that the co-chaperone serves as a brace interconnecting the other components of the complex (Fig. 1). The PPIase and TPR domains are connected by a 12-residue long hinge and adopt an open conformation without any direct inter-domain interaction (Fig. 5a). The interaction between AHR and XAP2 (1165 Å²) is mainly conferred through helix Fα of PAS-B (I338, I341, K342) and the carboxy-terminal extension (F404, F406, G409-E415, P420-I423) on the AHR side, while the XAP2 regions in contact with AHR include the loop

between strand βC′ and helix αII (K66-K69) within the PPIase domain, helix α2 and the loop α2-α3 of the TPR domain (Y203, I206, A207, M214-P218, W223), and the linker region (W168) (Fig. 3e). Single amino acid substitutions of three AHR residues at the interface with XAP2 (F404A, F406A, Y414A) had a deleterious effect on XAP2 binding without drastically altering Hsp90 interaction (Fig. 3f and Supplementary Fig. 6b).

XAP2 also forms an extensive contact with Hsp90 (1027 Å²) mainly via the carboxy-terminal domain of the Hsp90B and the carboxy-terminal MEEVD motif, the latter previously described[24] (Fig. 3g and Fig. 5b, c). We mutated XAP2 residues at the interface with Hsp90 (Y248A in α4, W279A in loop α5-α6, or D317A, D320A, F324A, I327A, F328A, S329A in helix α7). All mutations but D320A and S329A destabilised the Hsp90-XAP2 interaction (to various extents), but none of them considerably altered the interaction between AHR and Hsp90 (Fig. 3h and Supplementary Fig. 6c). Altogether, these data suggest that Hsp90 and AHR form a stable binary complex, independently of the presence or absence of XAP2.

**Exploring AHR ligand-binding mechanism**
Indirubin (Fig. 6a) is a dietary-derived endogenous ligand of AHR produced from tryptophan by intestinal bacteria[25]. We confirmed that indirubin is a potent activator of AHR using cell-based assay (Fig. 6b). We also demonstrated that the purified Hsp90-XAP2-AHR complex is active and binds indirubin, and that the presence of p23 is not required for the interaction with the ligand (Fig. 6c). In the cryo-EM structure, the indirubin binding site is unambiguously resolved within the PAS-B pocket (Fig. 6d). The AHR ligand-binding pocket (LBP) takes a form of an elongated channel extending perpendicularly between helices Cα and Fα (Fig. 6e). Remarkably, all elements of the PAS-B secondary structure contribute to the LBP. The amino acids lining the cavity are essentially hydrophobic (71%), including 8 aromatic (26%) and 14 aliphatic (45%) residues. The LBP also comprises 9 polar amino acids (29%) but is essentially devoid of charged residues (Supplementary Fig. 8a). The ligand occupies a volume of ~220 Å³ out of the total 682 Å³ of the void volume as determined by CASTp[26]. Accordingly, among 31 residues contributing to the LBP, only 15 side chains interact with indirubin (4.2 Å distance cut-off), indicating that a significant portion of the cavity remains unoccupied (Fig. 6e and Supplementary Fig. 8b). The amino acids involved in binding are located in the helix Fα region and involves residues originating mainly from Bβ, Eα, Fα, Gβ, Hβ and Iβ, whereas on the other side of the pocket, the volume delineated by Aβ, Cα, Dα and the loop Gβ-Hβ remains unoccupied (Fig. 6f and Supplementary Fig. 8c and Supplementary Movie 3). Indirubin is a planar molecule with an asymmetric double indole structure. The indole ring adjacent to helix Fα is engaged in many

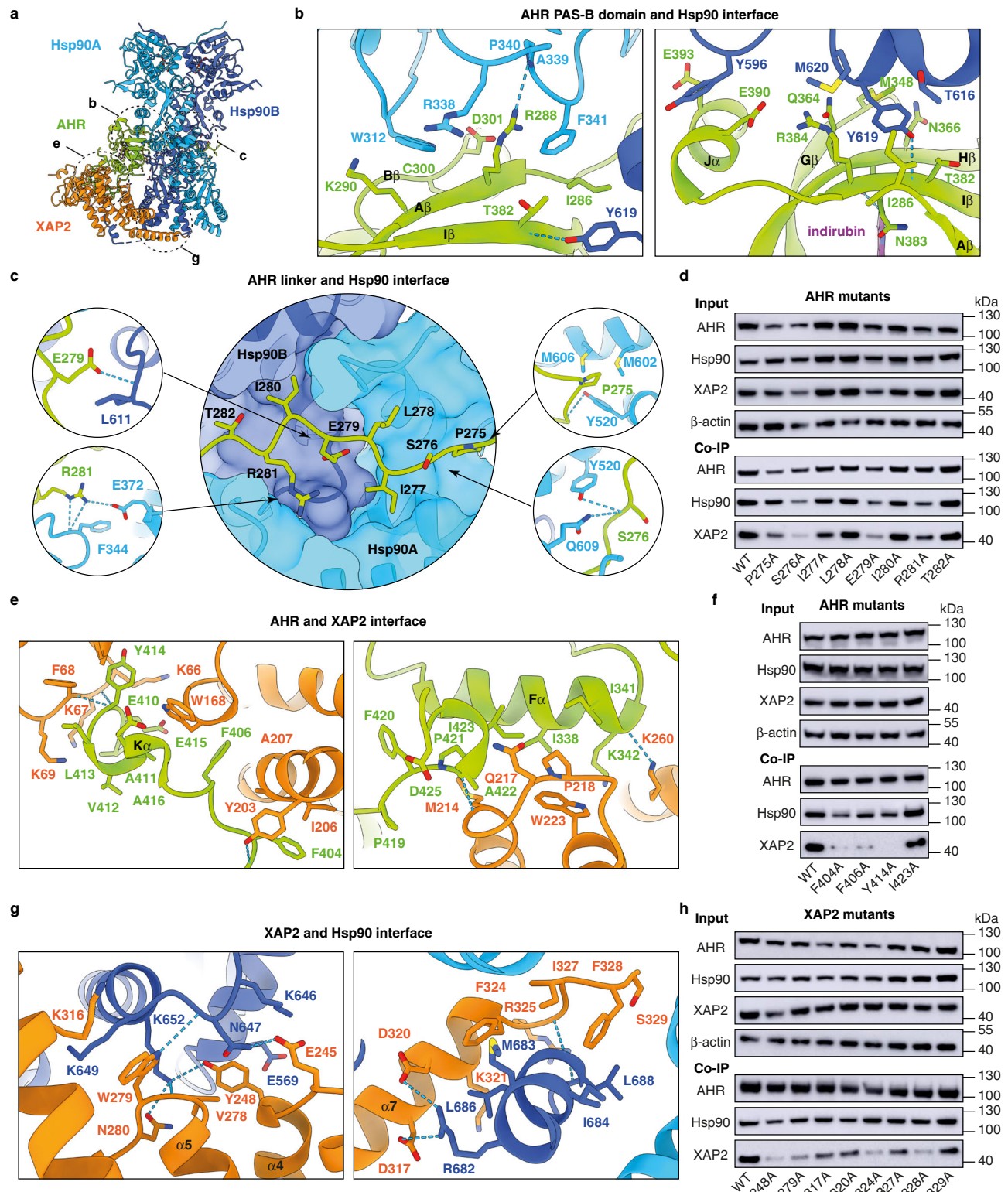

**Fig. 3 | Details of the Hsp90-XAP2-AHR complex organisation. a** An overall view of the complex with indicated location of the interaction sites between the proteins. The dashed circles indicate the contacts described in the following panels. **b** Two principal interaction sites between Hsp90 and PAS-B domains. **c** Close-up view of the AHR linker that is threaded through the Hsp90 lumen and interacts with both molecules. The residues for which we observed the strongest effect of the mutation on the complex stability in Co-IP assay are shown in small circles.

**d** Results of Co-IP for the mutation of the AHR linker. The data were representative of three independent experiments. **e** Close-up view of the interaction sites between AHR and XAP2. **f** Co-IP results for the AHR mutants at the interface with XAP2. The data were representative of four independent experiments. **g** Interactions between XAP2 and Hsp90. **h** Co-IP results for the XAP2 mutants at the interface with Hsp90. The data were representative of two independent experiments. Source data are provided as a Source Data file.

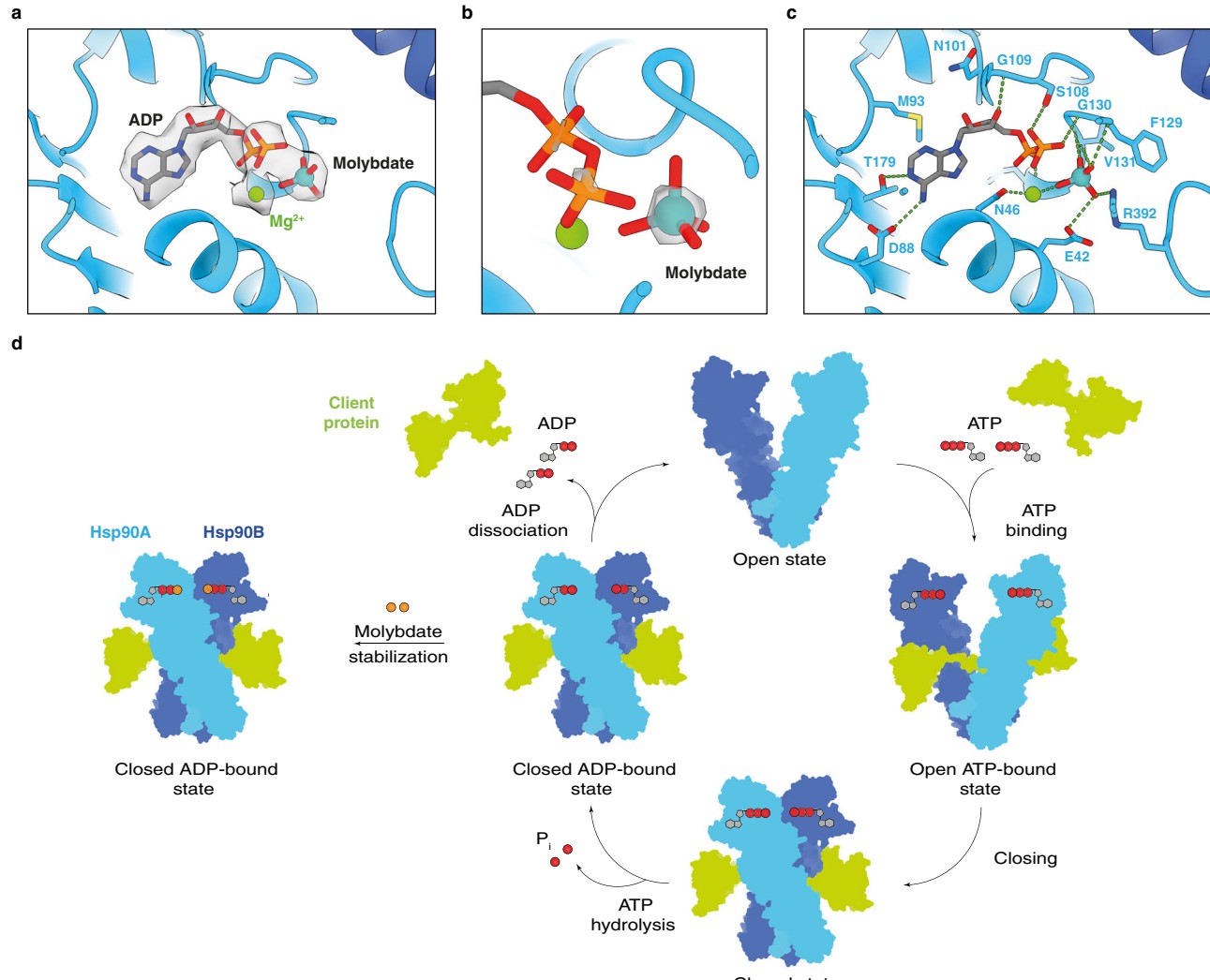

**Fig. 4 | Detailed analysis of the nucleotide binding site of Hsp90. a** Close-up view of the nucleotide binding site in molecule A of Hsp90. The electron density map for the molecule of ADP, molybdate anion and magnesium cation has been displayed. **b** Close-up view of the molybdate ion. Increased contour level of the map density shows a much stronger density relative to the α and β-phosphates. **c** Details of the organisation of the nucleotide binding site. Amino acid residues involved in the interactions are shown as sticks and labelled. Hydrogen bonds are indicated as dashed green lines. **d** Schematic representation of the conformational cycle of Hsp90. Binding and release of a client protein is coupled with binding and hydrolysis of ATP. Molybdate mimics the presence of ATP and stabilises a closed conformation of Hsp90 even in the presence of ADP.

contacts with aliphatic and aromatic pocket residues, namely F295, Y322, I325, C333, H337, I349 and F351. The second indole ring is primarily involved in two hydrogen bonds with S365 and Q383 via its carbonyl and amine moieties, respectively, in addition to several van der Waals contacts with H291, P297, G321, F324, F351, L353 and V381. The hallmark of most AHR ligands is their planarity. Our structure reveals that indirubin intercalates between H291, F295 and P297 on one side of the ligand and I325, C333, F351 and L353 on the other side, the two layers of residues exerting a critical role in the selection of indirubin, and most likely other AHR ligands, based on its planar structure (Fig. 6g).

Multiple microsecond-long molecular dynamics (MD) simulations were carried out to obtain an estimation of the relative energetic contributions of the LBP amino acids to the interaction between indirubin and the PAS-B domain. The ligand remains stable in the experimental pose and its presence stabilises the bound conformation of the protein (Supplementary Fig. 9a), maintaining stable contacts with most of the residues observed in the cryo-EM structure (Supplementary Fig. 9b). Most of these amino acids form favourable Lennard-Jones interactions with indirubin (Fig. 6h, upper panel), in particular

three residues with aromatic side chains form stable stacked or T-shaped π-interactions with the indole rings of indirubin, namely H291, F295, and Y322 (Fig. 6g and Supplementary Fig. 9c, d). Our MD simulations confirm the substantial implication of S365 and Q383 that are involved in stable hydrogen bonds with the ligand (Fig. 6h, lower panel). Interestingly, the presence of indirubin reduces the overall mobility of the backbone, in particular helices Cα, Dα, and the loop Gβ-Hβ (Fig. 6i). The high dynamics of these two neighbouring regions in the AHR apo form suggests a possible entry site for the ligand (Fig. 6j and Supplementary Fig. 10a, b). Note that MD simulations carried out using the whole complex instead of the isolated AHR PAS-B domain led to similar conclusions (Supplementary Fig. 10c–e, Supplementary Table 2 and 3; see Materials & Methods section for details).

To experimentally validate the indirubin binding mode, we next mutated several interacting residues and tested the binding properties of the purified mutant proteins using nano-DSF (Supplementary Fig. 11a). We observed that substitution of H291, F295 and S365 by alanine residue strongly reduced interaction with indirubin. Previous reports revealed significant species differences in AHR response to indirubin treatment[27]. Accordingly, we found that indirubin activates

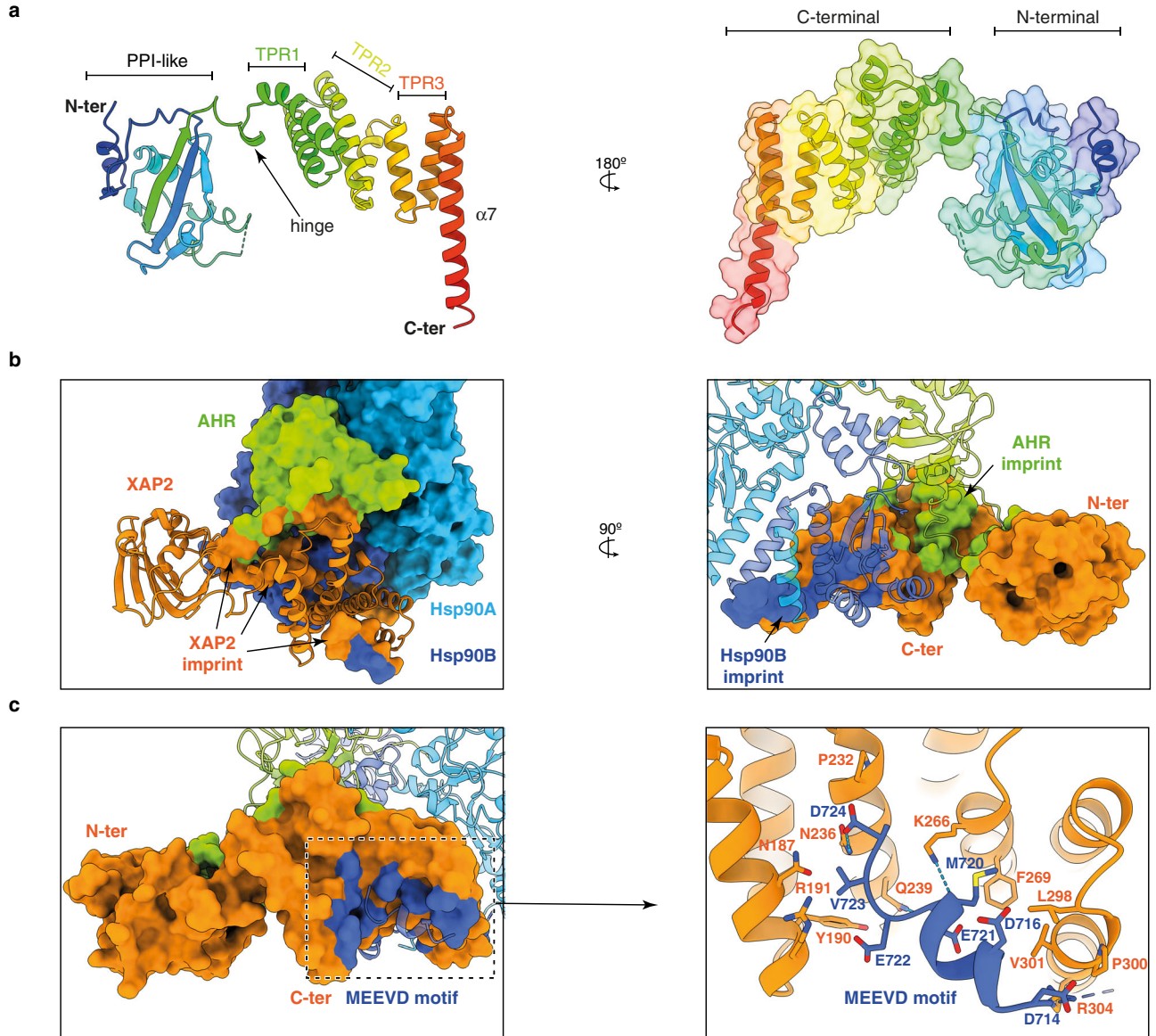

**Fig. 5 | Detailed analysis of XAP2 protein and its interactions with the other partners. a** Cartoon representation of XAP2 protein in two views coloured in rainbow from N- to C-terminus. Location of the most important features is indicated. XAP2 is composed of two domains that are separated by a short hinge. The N-terminal domain is not very well defined in the cryo-EM map most likely due to its increased mobility. The alpha-helical C-terminal domain is forming extensive contacts with both AHR and Hsp90 and is much better defined. PPI-like−peptidyl-prolyl isomerase-like domain. TPR1, 2 and 3−tetratricopeptide repeat motif 1, 2 and 3, α7−C-terminal helix alpha. **b** XAP2 is interacting extensively with two other proteins. Left panel: Imprint of the XAP2 on the other components of the complex. AHR and Hsp90 are shown in surface and XAP2 in cartoon representation; the AHR and Hsp90 residues in contact with XAP2 are coloured in orange. Right panel: Imprint of Hsp90 and AHR on XAP2. XAP2 is shown in surface, AHR and Hsp90 in cartoon representation. The XAP2 residues in contact with AHR and Hsp90 are coloured in green and blue, respectively. **c** Left panel: Imprint of the C-terminal part of Hsp90 on XAP2. Residues in contact with the MEEVD motif are coloured in dark blue. Right panel: Details of the XAP2 interactions with MEEVD motif of Hsp90. The C-terminal part of Hsp90 forms a helix that interacts with the TPR motifs. Hydrogen bonds are indicated as dashed blue lines.

the human receptor more potently than the rat and zebrafish homologues (Supplementary Fig. 11b). In full support of these data, sequence (Supplementary Fig. 11c) and structural (Supplementary Fig. 11d) analyses revealed that important stabilising interactions provided by I349 and V381 in human AHR are lost in the two other species which contain smaller residues at these positions (T and A, respectively). In the fish receptor, additional replacement of human S365 by an alanine residue is also very likely to explain a decrease in indirubin response due to removal of a stabilising hydrogen bond with the ligand.

Together, these observations support the notion that the indirubin binding site adjacent to helix Fα is the primary anchor point of AHR ligands (Fig. 6e). This region of the LBP harbours all structural and molecular determinants controlling ligand-binding specificity, promiscuity and affinity by imposing strong geometrical constraints in order to fit flat hydrophobic ligands (e.g., PAHs, HAHs, etc.), whilst providing polar amino acid residues with the potential to form hydrogen bonds with hydrophilic chemical groups (e.g., indole derivatives such as indirubin). Moreover, our cryo-EM structure implies how AHR can also accommodate larger compounds. The binding pocket extension towards helix Cα, referred to as the secondary binding site (Fig. 6e), appears to be less geometrically constrained and contains a mix of aliphatic and polar side chains that can engage in diverse types of contacts with compounds featuring higher three-dimensional hindrance.

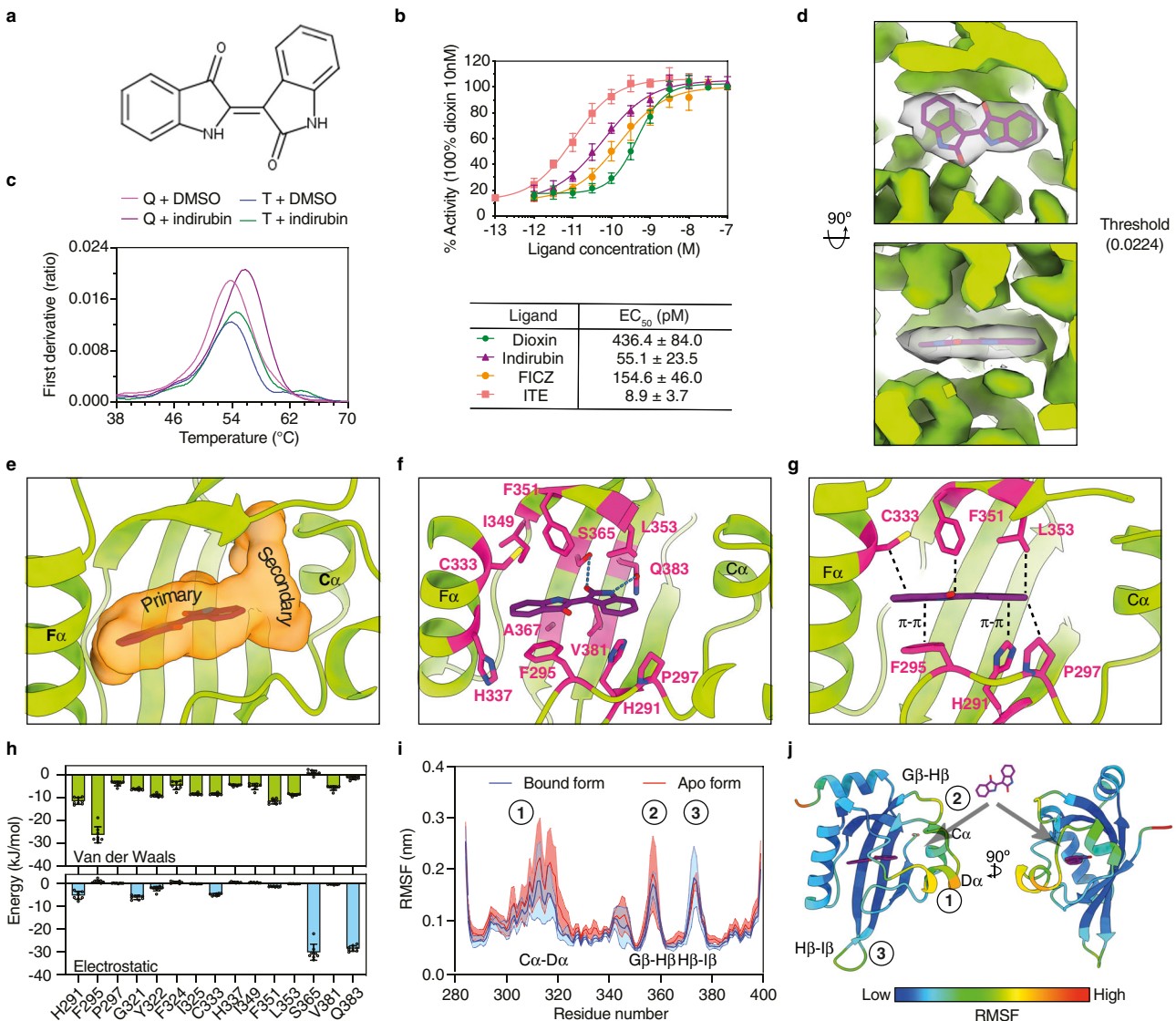

**Fig. 6 | Characterization of the interaction with indirubin. a** Structure of indirubin. **b** Results of cell-based activity assays for four AHR ligands. The results are represented as mean value ± SD from independent biological replicates (n = 4). **c** Nano-DSF analysis of the interaction between Hsp90-XAP2-AHR (T, ternary) and Hsp90-XAP2-p23-AHR (Q, quaternary) complexes and indirubin. The results are representative of independent biological replicates (n = 3). **d** Two orthogonal views of the cryo-EM map for the region corresponding to indirubin. **e** Close-up view of the ligand binding pocket (LBP) shown as an orange surface. **f** Close-up view of the indirubin binding site. Residues interacting with the ligand are shown as magenta sticks. Hydrogen bonds are indicated as dashed blue lines. **g** Two layers of LBP

residues determine AHR ligand selectivity. **h** Van der Waals (upper panel) and electrostatic (lower panel) interaction energies between indirubin and the residues of the protein. Data are presented as mean ± standard deviation of MD simulation replicas (n = 8). **i** Root mean square fluctuations (RMSF) of the protein backbone in bound (blue) and apo (red) conformations. Values are averages over the MD simulations; shaded areas represent the standard deviation between the replicas (n = 10 and n = 8 for the apo and bound system, respectively). **j** Two views of the PAS-B domain coloured according to the RMSF. The location of a potential ligand entry site to LBP is indicated with arrows. Source data are provided as a Source Data file.

## Discussion

Attempts to express, purify and solve the structure of the AHR ligand-binding domain, either in the context of the full-length protein or as an isolated domain have failed for many years. In our approach we reasoned that the reconstitution of the cytosolic complex by co-expressing AHR with its main chaperone and co-chaperone would help structurally and functionally stabilise the receptor. Using the Sf9 insect cell expression system we were able to obtain a stable protein sample and solve the cryo-EM structure of the indirubin-bound Hsp90-XAP2-AHR complex, at a resolution of 2.85 Å. This study provides structural insight into the details of the complex assembly. AHR is forming extensive interaction sites with both protein partners, partially explaining why all previous efforts to obtain the isolated PAS-B domain of human AHR have failed. The structure reveals how Hsp90

and XAP2 are interacting with AHR and maintaining the protein in the active form. The interactions involve not only the linker interconnecting AHR PAS-A and B domains and PAS-B itself but also the mainly disordered C-terminal extension of the protein. Along these lines, our structural and mutational analyses show that AHR and Hsp90 form a stable binary complex and suggest that XAP2 could play the role of a scaffolding protein stabilising the structural integrity of the client protein, notably the unstructured transactivation domain of AHR. Our cryo-EM structure of the complex also provides insight into a general mechanism of Hsp90-client recognition. Reminiscent of the GR and Cdk4 structures[18,20], the AHR protein is threaded through the Hsp90 dimer lumen, suggesting that the mechanism of client recognition is conserved between different client proteins. Particularly, the position of key hydrophobic residues within the extended linker region that

interact with Hsp90 seems to be conserved across all three available to date structures.

The structure explains previous observations suggesting that AHR Hsp90-binding and ligand-binding sites overlap[19]. Indeed, residues localised within PAS-B strands Aβ and Bβ are involved in the interactions with Hsp90 and the ligand. The proposed mechanism of AHR activation includes ligand binding and induction of conformational changes in AHR that could be directly sensed by Hsp90. Subsequently, the signal would be transferred to the amino-terminal part of the receptor leading to exposure of the NLS and triggering translocation to the nucleus. In case of AHR, the NLS sequence is predicted to be localised within the first 60 amino acid residues. Interestingly, as our complex was prepared in the presence of a ligand, the PAS-A domain and the N-terminal part of AHR could not be convincingly placed in the electron density suggesting high flexibility of this region. This part of the protein is therefore exposed to the solvent and can interact freely with the nuclear transport machinery including the importin complex.

One of the hallmarks of AHR is its remarkable promiscuity as the protein can bind many different ligands. Our cryo-EM structure allowed us to explain the mechanism behind selectivity towards certain types of ligands. Amino acid residues involved in the formation of the primary binding site are responsible for the selectivity towards molecules exhibiting a planar shape. Importantly, mutational analyses confirmed the importance of residues involved in the formation of π-interactions with the ligand. The structure also provides the explanation for receptor promiscuity. The ligand-binding pocket with its relatively large interior cavity can accommodate molecules of different sizes. Additionally, MD simulations provide insight into the potential entry site of the ligand. The region localised between helices Cα and Dα, exposed to the solvent and not obscured by the other components of the complex, exhibits an elevated degree of flexibility in the simulations. It is also worth noting that despite striking similarity in the overall structure, the PAS-A domain of AHR does not possess any internal cavity that could serve as a potential binding pocket. Instead, the interior of the domain is occluded by a presence of several bulky aromatic amino acid residues supporting the hypothesis that in the course of its evolution the PAS-A domain has been selected to serve primarily in protein-protein interactions.

Interestingly, previous comparison studies between human and rodent proteins also revealed intriguing interspecies differences concerning AHR specificity. Despite relatively high sequence homology between human and mouse AHR receptors (around 85%), the same chemical molecule can interact with the orthologous proteins with drastically different affinities. For example, mouse AHR binds dioxin with a tenfold higher affinity than its human counterpart[28]. In contrast, human AHR binds indirubin and indoxyl sulphate with much higher affinity compared with the mouse receptor[29,30]. Intriguingly, human AHR protein appears to have evolved towards better recognition of compounds belonging to the indole family, which is most likely directly connected with its involvement in the signalling in the human digestive track[31]. Despite high promiscuity of the PAS-B domain, its specificity can still be finely tuned by small variations in the protein sequence. Indeed, our analyses of AHR polymorphism between human, rat and zebrafish combined with structural data explain the interspecies differences observed in the binding affinity for indirubin. Our structure demonstrates how microbiome-derived compounds can interact with a human protein and modulate host signalling pathways. Therefore, it provides an insight into the molecular mechanism that emerged as a result of coevolution between the human body and its microbiome.

By disclosing the atomic model for the AHR cytosolic complex and for its PAS-B domain bound to a ligand, our structure has enabled us to explain many prior biochemical observations. Moreover, it provides a strong rationale for future mechanistic studies and development of novel pharmaceuticals that could find applications in the treatment of many diseases, including irritable bowel disease, allergic responses and cancer.

## Methods

No statistical methods were used to predetermine sample size. The experiments were not randomised, and investigators were not blinded to allocation during experiments and outcome assessment.

### Preparation of DNA constructs for structural studies

Synthetic DNA coding for human aryl hydrocarbon receptor (AHR, UniProt accession number P35869), heat shock protein HSP 90-beta (Hsp90, UniProt accession number P08238), HBV X-associated protein 2/aryl-hydrocarbon receptor-interacting protein (XAP2/AIP, UniProt accession number O00170) and prostaglandin E synthase 3 (p23, UniProt accession number Q15185) was ordered from Integrated DNA Technologies. AHR encompassing residues 1-437 was fused at its N-terminus with Twin Strep-tag and maltose binding protein (MBP). Tobacco Etch virus NIa protease (TEV) cleavage site was introduced between AHR and the tag portion of the construct. Full length Hsp90 was fused with $His_6$-tag at its N-terminus. Full length XAP2 and p23 were left untagged. The DNA was cloned into pBAC4x-1 plasmid (Novagen) using In-Fusion kit (Takara Bio) and standard molecular biology techniques. The results were verified by sequencing (GENEWIZ). The final DNA construct allows expression of all 4 proteins simultaneously: Twin Strep-tag-MBP-TEV site-$AHR_{1-437}$, $His_6$-tag-$Hsp90_{1-724}$, $XAP2_{1-330}$ and $p23_{1-160}$. As the endogenous Hsp90 protein present in the insect cells is also able to form complexes with human AHR, double affinity purification strategy was necessary to assure that human Hsp90 exclusively will be present in the final sample. For binding studies, point mutations (H291A, F295A, S365A and Q383A) were introduced within the PAS-B domain of AHR using QuikChange Lightning Site-Directed Mutagenesis Kit (Agilent) and the results were confirmed by sequencing (GENEWIZ). The DNA sequences of the oligonucleotides used to introduce the mutations are provided in the Supplementary Table 4.

### Protein expression and purification

*Spodoptera frugiperda* Sf9 cells (Oxford Expression Technologies) were co-transfected using the described above plasmid construct and flashBAC ULTRA viral DNA (Oxford Expression Technologies). Protein expression was induced by inoculation of the cell culture at around $4 \times 10^6$ cells/ml with the second passage stock (P2) of the virus. 36 h post-infection cells were harvested by centrifugation, resuspended in the freezing buffer containing 50 mM Bis-Tris-HCl pH 7.0, 50 mM NaCl, 10 mM KCl, 10 mM $MgCl_2$, 20 mM $Na_2MoO_4$ 20 mM imidazole, and 2 mM 2-mercaptoethanol (BME) and immediately used for protein purification without storing.

For protein purification, cell pellets were resuspended in the freezing buffer supplemented with 2 mM PMSF (Sigma-Aldrich) and one tablet of cOmplete™ EDTA-free Protease Inhibitor Cocktail (Roche). Cells were lysed on ice using sonication and centrifuged at 40,000 x $g$ for 30 min at 8 °C. The supernatant was filtered through 0.45 μM syringe filter (Sartorius) and applied onto 5 ml HisTrap HP column (Cytiva). The column was extensively washed using 50 CV of the washing buffer containing 20 mM Bis-Tris-HCl pH 7.0, 50 mM NaCl, 10 mM KCl, 10 mM $MgCl_2$, 20 mM $Na_2MoO_4$, 20 mM imidazole and 2 mM BME. The protein complexes were eluted using a washing buffer supplemented with 500 mM imidazole. The eluted fractions were subsequently applied onto 5 ml Strep-Tactin®XT Superflow® column (IBA Lifesciences) preequilibrated with 20 mM Bis-Tris-HCl pH 7.0, 50 mM NaCl, 10 mM KCl, 10 mM $MgCl_2$, 20 mM $Na_2MoO_4$ and 2 mM BME. The protein was eluted with 50 mM biotin and incubated overnight with the in-house produced TEV protease in order to cleave off the Twin Strep-tag-MBP. The digestion solution was also supplemented with 2 mM ATP-$MgCl_2$ (Sigma-Aldrich) to stabilise the closed

conformation of Hsp90. On the following day, the sample was filtered through 0.2 μM syringe filter (Sartorius) and loaded onto Capto HiRes Q 5/50 column (Cityva). The unbound material was washed away with 20 mM Bis-Tris-HCl pH 7.0, 50 mM NaCl, 10 mM KCl, 10 mM MgCl$_2$, 20 mM Na$_2$MoO$_4$ and 2 mM BME and the protein complexes with and without p23 were separated using 20 CV of 0-500 mM NaCl gradient. At this stage, the fractions containing either a ternary complex Hsp90-XAP2-AHR or a quaternary complex Hsp90-XAP2-p23-AHR were pooled separately and diluted 1:3 in the buffer containing 20 mM Bis-Tris-HCl pH 7.0, 10 mM KCl, 10 mM MgCl$_2$, 20 mM Na$_2$MoO$_4$, 2 mM ATP-MgCl$_2$ and 2 mM BME. The sample was then concentrated using Amicon® Ultra-4 with the molecular weight cut-off 100 kDa (Millipore) and injected onto Superdex® 200 Increase 10/300 GL column (Cityva) preequilibrated with the final buffer containing 20 mM Bis-Tris-HCl pH 7.0, 50 mM NaCl, 10 mM KCl, 10 mM MgCl$_2$, 20 mM Na$_2$MoO$_4$ and 2 mM BME. All chromatography steps were performed using Äkta Pure protein purification system (Cityva) operating at 4 °C. The 0.5 ml protein fractions were used immediately for grid preparation or concentrated using Amicon® Ultra-4 with the molecular weight cut-off 100 kDa (Millipore), flash-frozen in liquid nitrogen and stored at −80 °C. The expression and purification of mutant proteins were performed in the analogical way. Protein concentration was calculated using absorption coefficient A$_{280}$ 0.1% = 0.756. The absorption coefficients were obtained using the programme ProtParam from the ExPASy server (http://web.expasy.org/protparam/).

### Preparation of the complex with indirubin
Indirubin was purchased from Sigma-Aldrich (catalogue number SML0280). The compound was dissolved at 20 μM in DMSO (Sigma-Aldrich) and added to the protein solution at a 5x molar excess and 0.25% (v/v) final DMSO concentration. Before preparation of the grids, the complexes were incubated on ice for 1 h to allow sufficient time for binding.

### Sample preparation for cryo-EM and data collection
The grids were prepared within 2 h following the final step of the complex purification. Only the highest concentration fraction from the middle of the size exclusion peak was used. Using the same buffer as used for the size exclusion chromatography, the proteins were diluted to the final concentration of 0.18-0.20 mg/ml. Typically, 3 μl of the protein solution were applied on the C-Flat CF-1.2/1.3-3Au grids (Protochips) that were glow-discharged using PELCO easiGlow™ Glow Discharge Unit (Ted Pella, Inc.). The grids were then blotted for 2.5 until 4.5 s at 100% humidity and room temperature and plunge-frozen in liquid ethane using FEI Vitrobot (Thermo Fisher Scientific). Grids were stored in liquid nitrogen until shipment and data collection.

Cryo-EM data were collected at The Midlands Regional Cryo-EM Facility in Leicester, UK using FEI Titan Krios transmission electron microscope (Thermo Fisher Scientific) operating at 300 kV and equipped with Gatan K3 direct electron detector camera (Gatan) and GIF Quantum energy filter (Gatan) set to a slit width of 20 eV. EPU v.2.11 software was used for automatic data collection. Images were collected at nominal magnification of 81,000 in a super-resolution counting mode with a calibrated pixel size of 1.086 Å and an accumulative dose of 15 e-/pix/s and 43 frames per movie. The applied defocus range varied between −2.7 and −1.5 in 0.3 μm intervals. Details of the data collection parameters are presented in Supplementary Table 1.

### Cryo-EM data processing
Data processing was performed using RELION3.0 and 3.1[32]. The movies frames were gain-corrected, drift-corrected, dose-weighted, aligned and motion-corrected using MotionCor2[33]. The initial contrast transfer function (CTF) values for each micrograph were estimated using CTFFIND4.1[34] after which the poor-quality images were discarded. The

reference-free particle picking was done with a Laplacian-of-Gaussian auto-picking mode in RELION. The detailed processing pipeline is depicted in Supplementary Fig. 2. A total of 11,546,649 particles were picked out of 8,900 selected images. The particles were initially binned by 2 and extracted with a box size of 120$^2$ pixels. After one round of 2D classification, a subset of 11,343,679 particles was selected and subjected to 3D classification with 4 sub-classes, regularisation parameter $T = 4$ and with an initial model generated with the same set of particles and a low-pass filtered to 20 Å resolution. Among the four 3D classes, a dominant class containing 3,486,142 particles (30.7%) and displaying clear features of secondary structural elements was selected for further steps. The particles were then subjected to a 3D refinement step that yielded four sub-classes. One sub-class was identified as a dimer of Hsp90 protein alone. The second sub-class consisted of defective Hsp90-AHR complex particles without the XAP2 co-chaperone. The two remaining sub-classes containing a total of 942,224 particles of the Hsp90-XAP2-AHR ternary complex were selected and subjected to another round of 3D classification. All eight sub-classes displayed high-resolution features and the selected 1,655,792 particles were unbinned, extracted with a box size of 320$^2$ pixels and refined yielding a 3.90 Å reconstruction. The particles were then subjected to 3D refinement without alignment that yielded three sub-classes. Out of them, two high-resolution sub-classes containing 678,724 particles were combined and subjected to 3D refinement followed by per-particle CTF refinement and dose-weighting yielding a final global map at 2.85 Å resolution.

To improve the resolution of the XAP2 and C-terminal part of AHR we created a mask around AHR PAS-B and XAP2 portion of the complex and using unbinned data we performed a focused refinement with signal subtraction. Multiple 3D classifications with number of classes ($K = 2$, 3, and 4) and values of Tau ($T = 10$, 20, 30 and 40) were carried out followed by 3D refinement for each obtained class. The particles belonging to the highest resolution class were selected and subjected to per-particle CTF refinement and dose-weighting yielding a focused map at 4.07 Å resolution. The overall resolution of each reconstruction was estimated using gold-standard Fourier shell correlation (FSC) = 0.143 criterion between the two half-maps[35,36] and presented in Supplementary Fig. 3f, h. FSC curves were calculated with a soft mask using Phenix Mtriage[37]. The composite map shown in Fig. 1a was obtained using the 'vop add' function in Chimera. Sphericity values for the electron density maps displayed in Supplementary Table 1 were calculated using the 3D FSC server (https://3dfsc.salk.edu)[38].

### Model building and refinement
Initial steps of model building were performed using Chimera[39]. For Hsp90 the protein model was based on the cryo-EM structure of the Hsp90-Cdk4-Cdc37 complex (PDB access code 5FWP). The initial model for a PAS-B domain of AHR was generated using I-Tasser[40] with CLOCK-BMAL1 (PDB access code 4F3L) and HIF-α-ARNT (PDB access code 4ZPR) structures as templates. The missing C-terminal part of the protein was built manually in Coot v0.8.9.2[41]. For the XAP2 protein, the initial model was assembled from the solution structure of the PPIase domain (PDB access code 2LKN) and crystal structure of PR domain in complex with human Hsp90 peptide (PDB access code 4AIF). The hinge region interconnecting the two domains was reconstructed manually. The focused map was used to build and refine the model for XAP2 protein and C-terminal part of AHR that was subsequently merged with the model obtained from the global map. The model of the PAS-B domain was build based on the global map that displayed higher data quality for this region compared with the focused map. The model of indirubin and associated constraints file were generated using Grade (http://grade.globalphasing.org) and placed inside the PAS-B domain. The asymmetric shape of the indirubin molecule was helpful with a correct placement of the model within the electron density. The structure was refined using Phenix

Refine[37]. FSCs model versus map were calculated using Phenix. The geometry of the final structure was validated using Molprobity[42].

## Native mass spectrometry

Prior to MS analysis, the protein complex was buffer exchanged into 100 mM ammonium acetate buffer pH 7.0 (Sigma-Aldrich) using Bio-Spin microcentrifuge columns (Bio-Rad Laboratories). Intact MS spectra were recorded on a Synapt G2-Si HDMS instrument (Waters Corporation) modified for high mass analysis and operated in ToF mode. Samples were introduced into the ion source using borosilicate emitters (Thermo Fisher Scientific). Optimised instrument parameters were as follows: capillary voltage 1.4 kV, sampling cone voltage 150 V, offset voltage 120 V, trap collision energy 100, transfer collision voltage 25 V, argon flow rate 8 ml/min and trap bias 5 V. Data were processed using MassLynx v.4.2 (Waters).

## Analytical mass spectrometry and detection of molybdate

Samples of the complex prepared as described in the 'Protein expression and purification' paragraph were injected onto Superdex® 200 Increase 10/300 GL column (Cityva) preequilibrated with the buffer devoid of molybdate: 20 mM Bis-Tris-HCl pH 7.0, 50 mM NaCl, 10 mM KCl, 10 mM MgCl$_2$ and 2 mM BME. The separation was performed using Äkta Pure protein purification system (Cityva) operating at 4 °C. The 0.5 ml protein fractions were pooled, concentrated using Amicon® Ultra-4 with the molecular weight cut-off 100 kDa (Millipore), flash-frozen in liquid nitrogen and stored at −80 °C. For the analysis, 100 μl of sample were diluted in 1% HNO$_3$ (v/v) in order to obtain a final solution of 10 ml. Trace element concentrations were determined with the Thermo Scientific iCAP TQ ICP-MS (using the Kinetic Energy Discrimination mode and He as collision gas on the "Plateforme AETE-ISO, OSU OREME, Université de Montpellier-France"). An internal solution, containing Be, Sc, Ge, Rh and Ir was added on-line to the samples to correct signal drifts. A calibration curve including four points (0, 1, 5 and 10 ppb) was analysed every 20–30 samples. The quality of the Mo analysis was checked by analysing international certified reference waters (CNRC SLRS-6). The accuracy was better than 10% relative to the certified values and the analytical error (relative standard deviation) was better than 3%.

## Immunoprecipitation assay

The DNA was cloned into pcDNA3 (Invitrogen) using In-Fusion kit (Takara Bio) and standard molecular biology techniques. The final constructs are as follows: Twin Strep-tag-MBP-TEV site-AHR$_{1-437}$, His$_{10}$-tag-FLAG-tag-TEV site-Hsp90$_{1-724}$ and Myc-tag-XAP2$_{1-330}$. Point mutations were introduced using QuikChange Lightning Site-Directed Mutagenesis Kit (Agilent) and the results were verified by sequencing (GENEWIZ). The DNA sequences of the oligonucleotides used to introduce mutations are provided in the Supplementary Table 4.

For a routine cell culture, HEK293F suspension-adapted cells (FreeStyle293F cells, Thermo Fisher Scientific) were grown and maintained in 30 ml free-styleTM 293 expression media (Gibco) in 125 ml cell culture flasks (Corning) according to manufacturer's protocol in an orbital shaker incubator at 37 °C, 140 rpm, and 8% CO$_2$. For the transfection assays, cells were seeded at $0.8 \times 10^6$ cells/ml into a final volume of 20 ml of the suspension medium per each 125 ml flask. A total of 20 μg of DNA were used to co-transfect the cells, corresponding to 13.3, 3.3, and 3.3 μg of expression vectors for AHR, Hsp90 and XAP2, respectively. 20 μg of DNA were added to 2 ml of the Dulbecco's phosphate buffered saline (DPBS, Gibco) and vortexed for 5 s. Subsequently, 40 μl of 0.5 mg/ml filter-sterilised Polyethylenimine (PEI, transfected agent, Sigma Aldrich) was added to the PBS/DNA solution, the mix was vortexed for 5 s and incubated at RT for 20 min. This DNA/PEI mix was added to the cells and incubated in an orbital shaker for a further 48 h at 37 °C, 140 rpm, and 8% CO$_2$. At the end of the transfection, $20 \times 10^6$ of cells were harvested at $250 \times g$ for 5 min.

The cell pellets were washed with 5 ml of DPBS and centrifuged at $1600 \times g$ for additional 5 min. The supernatant was carefully removed and the cell pellets were immediately frozen at −20 °C and stored until further processing.

All steps of the immunoprecipitation assay were performed on ice. Cell pellets were resuspended in 3 ml of the assay buffer containing 20 mM Bis-Tris-HCl pH 7.0, 10 mM KCl, 10 mM MgCl$_2$, 20 mM Na$_2$MoO$_4$ and 2 mM BME supplemented with cOmplete™ EDTA-free Protease Inhibitor Cocktail (Roche) and DNase I (Sigma-Aldrich). The samples were lysed by three cycles of repeated freezing and thawing. The obtained cell lysates were subsequently cleared by centrifugation at $5000\,x\,g$ for 10 min. 200 μl of the resulting supernatants were transferred into a new tube, mixed with 25 μl of 50% anti-MBP Magnetic beads (New England Biolabs) and agitated for 1 h. The beads were washed three times with 20 beads volume of the assay buffer and eluted with 100 μl of the 2x SDS PAGE blue buffer using heat denaturation for 5 min at 95 °C. Samples were then analysed using Western blot technique. For the input control, 250 μl of supernatant were mixed with 250 μl of sample buffer and heated for 10 min at 95 °C. The sample were centrifuged at $11,000\,x\,g$ for 1 min and resolved using 4–12% Bis-Tris Plus SDS-PAGE gels (Invitrogen). The proteins were transferred to the PVDF membrane (Bio-Rad TransBlot Turbo Midi) using the Trans-Blot Turbo Transfer System (Bio-Rad). The membranes were incubated in 5% (w/v) skimmed milk resuspended in DPBS (Sigma-Aldrich) with agitation for 1 h at 4 °C. The primary antibodies were then added and the membranes were incubated with agitation overnight at 4 °C. The primary antibodies used in this study: rat anti-MBP, (Sigma-Aldrich), dilution 1:1,000, rat anti-DYKDDDDK clone L5 (anti-FLAG, BioLegend), dilution 1:1,000, mouse anti-beta-actin (Proteintech), dilution 1:5,000, rat anti-Myc-tag clone number [9E10] (ABCAM), dilution 1:1,000 and rat anti-Strep-tag clone number [11A7] (ABCAM), dilution 1:2,000. The following day, the membranes were washed once with DPBS supplemented with 1% (v/v) Tween 20 and twice with DPBS. The secondary antibodies were then added, the membranes were agitated for 1 h at 4 °C and the wash step was repeated. The secondary antibodies used in this study: sheep anti-mouse HRP conjugated (GE Healthcare), dilution 1:3,000 and goat anti-rat HRP conjugated clone number poly4054 (BioLegend), dilution 1:5000. Finally, the membranes were revealed using ECL SuperSignal West Blot PLUS Chemiluminescent substrate (Thermo Fisher Scientific) and scanned using AMERSHAM Imager 600. The uncropped scans of the Western blot membranes are provided in the Source Data file.

## Thermal stability measurements

Thermal stability of the complexes was analysed using a Tycho NT.6 (NanoTemper Technologies). The proteins were diluted to a final concentration of 2.5 μM into the buffer containing 20 mM Bis-Tris-HCl pH 7.0, 50 mM NaCl, 10 mM KCl, 10 mM MgCl$_2$, 20 mM Na$_2$MoO$_4$ and 2 mM BME. Indirubin was dissolved at 20 mM in DMSO and added to the samples to a final concentration of 12.5 μM. For the reference samples, DMSO alone was added to the final concentration 5% (v/v), the same as in the presence of the ligand. After addition of the ligand, samples were incubated at room temperature for 15 min. Thermal measurements were carried out in a range from 35 to 95 °C with steps 1 °C per min. The resulting melting curves were generated by plotting the first derivative of the fluorescence ratio at 330 nm/350 nm against temperature.

## Generation of AHR reporter cell lines and culture medium

Cell culture materials were obtained from Life Technologies (Cergy-Pontoise, France). Luciferin sodium salt was purchased from Promega (Charbonnières, France). Dioxin was purchased from Campro Scientific (Berlin, Germany). All other compounds used in this study were obtained from Sigma-Aldrich (Saint-Quentin Fallavier, France). Stock solutions of the compounds were prepared in DMSO and stored at

−20 °C. Fresh dilutions in the test medium were made up before each experiment. The final DMSO concentrations during treatment did not exceed 0.1% (v/v) of the test medium.

The reporter HAhLH cell line was obtained by transfecting human HeLa cells with the dioxin-responsive gene XRE(TnGCGTG)$_3$-tata-luciferase-Luc-hygromycin plasmid as described before[43]. H4AhLH and ZAhLH cell lines were obtained in a similar way by transfecting rat H4IIE and zebrafish ZFL cells with the dioxin-responsive gene XRE(TnGCGTG)$_3$-tata-luciferase-Luc-hygromycin plasmid. HAhLH and H4AhLH cells were grown in Dulbecco's Modified Eagle's Medium F12 (DMEM) with phenol red, supplemented with 5% foetal calf serum (FCS), 1% antibiotics (penicillin/streptomycin) and 0.25 mg/ml hygromycin (culture medium) in a 5% CO$_2$ humidified atmosphere at 37 °C. For the cell treatment with compounds, phenol red-free DMEM/F-12 medium supplemented with 5% dextran-coated charcoal (DCC)-treated FBS and 1% penicillin/streptomycin was used (test medium). ZAhLH cells were cultured at 28 °C in humidified atmosphere with 5% CO$_2$ in LDF medium (50% Leibovitz 15 culture medium L15, 35% DMEM high glucose and 15% Ham's-F12 medium) with 0.15 g/l sodium bicarbonate, 15 mM 4-(2-hydroxyethyl)−1-piperazineethanesulfonic acid (HEPES), 0.01 mg/ml insulin, 50 ng/ml epidermal growth factor (EGF), 50 U/ml penicillin and streptomycin antibiotics, 10% (v/v) foetal bovine serum (FBS). For cell treatment with the compounds, same culture medium was used except 10% (v/v) foetal bovine serum (FBS) was replaced by 5% of dextran-coated charcoal FBS (DCC).

### Reporter cell line assay
HAhLH, H4AhLH and ZAhLH reporter cells were seeded in 96-well white opaque tissue culture plates at a density of 50,000 cells per well in 150 μl of the test medium. Test compounds were prepared at 4x concentration in the same medium, and 50 μl per well were added 24 h after seeding. Cells were incubated for 8 h in the presence of the compounds at 37 °C (HAhLH, H4AhLH) or 28 °C (ZAhLH). At the end of incubation, the medium containing test compounds was removed and replaced with test culture medium containing 0.3 mM luciferin and luminescence was measured in intact living cells for 2 s. Experiments were performed in quadruplicate for each ligand concentration. Each compound was tested in at least four independent experiments. Results were expressed as a percentage of maximal luciferase activity. Maximal luciferase activity (100%) was obtained in the presence of 10 nM (HAhLH, H4AhLH) or 30 nM (ZAhLH) dioxin. Dose response curves were modelled using GraphPad Software (version 5.0 Inc., San Diego, CA). Effective concentrations (ECs) are derived from the Hill equation. For a given compound, EC$_{50}$ is defined as the concentration inducing 50% of its maximal effect. The EC$_{50}$ values were calculated including the adjustment for the basal activity of the cell line.

### Molecular dynamics simulations
All MD simulations were performed using Gromacs 2022[44]. The ff19sb force field[45] was employed to model the protein molecule, while ligands and water molecules were modelled with GAFF2[46] and OPC[47] force fields, respectively. Force field parameters were assigned by means of the tleap tool in AmberTools20[48] and were successively converted in Gromacs format using the amb2gro_top_gro.py script, also available in AmberTools20. The starting configurations for indirubin-bound and apo PAS-B domain (284-399) were taken directly from the cryo-EM structure of the complex.

In simulations of the full complex, missing loops of Hsp90 (residues 220-275) and XAP2 (residues 110-133) molecules were modelled using the graphical interface of MODELLER available in UCSF Chimera 1.14[49]. Parameters for the molybdate ions were adapted from non-bonded parameters of phosphate ions and molybdate structural data (see Supplementary Tables 2 and 3). The remarkable stability of the molybdate ions in the experimental binding pose during the MD runs (RMSD ≤ 0.1 nm) reassured us about the reliability of this approximation.

For the simulations of the apo system, indirubin was removed from the complex. Initial configurations were solvated in a rhombic dodecahedron box of 230 nm$^3$ for simulations of PAS-B domain, and of 4300 nm$^3$ for simulations of the full complex, neutralising the net charge of the system with 0.1 M of NaCl. A 10 ns simulation restraining the position of the heavy atoms of protein and ligand molecules was performed to let the water molecules equilibrate followed by 1 microsecond without restraints for each replica were performed in the NPT ensemble ($T = 300$ K, $P = 1$ atm), controlling temperature and pressure by means of the v-rescale[50] and Parrinello-Rahman[51] algorithms, respectively. Periodic boundary conditions were applied, and long-range electrostatic interactions were computed with the particle mesh Ewald method[52] with cutoff of 1 nm for real space interactions, while van der Waals interactions were computed with a cutoff distance of 1 nm. All bonds were constrained using LINCS[53], allowing for a timestep of 2 fs for the integration of the equations of motion. Lennard-Jones and electrostatic interaction energies between the protein residues and indirubin were extracted by means of the gmx energy tool in Gromacs 2022.

Root mean square fluctuation (RMSF) of the backbone of bound and apo conformations were computed with the gmx rmsf tool, averaging the values for each residue. The gmx rms tool was used to compute the root mean square deviation (RMSD) of the backbone atoms of the protein in bound and apo conformations and of the indirubin molecule with respect to the backbone of the protein. Minimum distances between the heavy atoms of the residues of the protein and indirubin were computed with the gmx pairdist tool and residues were counted as in contact with indirubin when this distance was below 0.42 nm. To evaluate stacking and T-shape conformations, the angles between the rings of indirubin and of the amino acids were computed using the gmx gangle tool, while the distance between the centre of geometry of the same rings was evaluated by means of the gmx pairdist tool.

### Data analysis and figure preparation
Figures were generated using UCSF Chimera v1.13.1[39] and ChimeraX v1.3[54]. Ligand binding data were analysed with GraphPad Software version 5.0 Inc., San Diego, CA. PAS-B topology diagram was obtained using Pro-origami[55]. The cavity within the PAS-B domain vas calculated using CASTp[26]. Sequence alignment was performed using Clustal Omega[56] and visualised using ESPript 3.0 (https://espript.ibcp.fr)[57]. Plots were generated using Prism v9 (GraphPad).

### Reporting summary
Further information on research design is available in the Nature Portfolio Reporting Summary linked to this article.

## Data availability
The data that support this study are available from the corresponding authors upon request. The atomic coordinates for the indirubin-bound Hsp90-XAP2-AHR complex have been deposited in the Protein Data Bank (PDB) under the accession code 7ZUB. The cryo-EM map obtained in this study has been deposited in the Electron Microscopy Data Bank (EMDB) under the accession code EMD-14971. Focused map used for model refinement has been deposited in EMDB under the accession code EMD-14972. Source data are provided with this paper.

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

## Acknowledgements

We acknowledge The Midlands Regional CryoEM Facility at the Leicester Institute of Structural and Chemical Biology (LISCB), major funding from MRC (MC_PC_17136). We thank Rémi Freydier and Léa Causse (AETE-ISO platform, OSU-OREME/Université de Montpellier) for the determination of molybdate concentrations in our samples. Centre de Biologie Structurale is a member of the French Infrastructure for Integrated Structural Biology (FRISBI), a national infrastructure supported by the French National Research Agency (ANR-10-INBS-04-01). Native MS experiments were carried out at the Montpellier Proteomics Platform (PPM, Bio-campus Montpellier) supported by the regional funds FEDER/Région Occitanie, MUSE and Labex EpiGenMed. The work was supported by funding from European Union's Horizon 2020 research and innovation programme grant GOLIATH No. 825489 (W.B.), ATIP-Avenir 2020 grant No. R20059SP (J.G.) and the ANR (the French National Research Agency) "Investissements d'avenir" programme reference No. ANR-16-IDEX-0006 (J.G.).

## Author contributions

J.G. and W.B. initiated and supervised project. J.G. and L.G. purified protein samples. J.L.-K.-H. and A.A. prepared EM grids. J.G., J.L.-K.-H., C.G.S., J.B. and A.A. collected and analysed EM data. J.G., H.-S.K. and W.B. performed model building and refinement. J.G. and W.B. analysed the structure. L.G. and P.G. carried out biochemical analysis. M.G., A.Bo. and P.B. performed cell-based assays. M.P. and A.Ba. performed molecular dynamics simulations. C.B. collected and analysed MS data. J.G. and W.B. principally wrote the manuscript which was finalised with input from all authors.

## Competing interests

The authors declare no competing interests.
