## [Peer Review File · Nature Communications]

Cryo-EM structure of the agonist-bound Hsp90-XAP2-AHR cytosolic complexREVIEWER COMMENTS

Reviewer #1 (Remarks to the Author):

In the manuscript, the 2.85 Å Cryo-EM structure of the Hsp90-XAP2-AhR complex, bound to the indirubin ligand, is presented and analyzed.

A number of noteworthy information related to the structure and interactions of this complex are revealed in this work. These structural evidences have represented unanswered questions in the research related to the ligand-activated mechanism of the Aryl hydrocarbon Receptor (AhR) for many years. Given the key regulatory role that this transcription factor plays in modulating the ability of diverse chemicals to produce a wide spectrum of biological and toxicological effects, combined with its identification as a significant target for the development of therapeutic drugs, an understanding of its molecular mechanism is of great relevance.

As reported in extensive well-established literature, due to the lack of experimental structural information on the PAS-B ligand binding domain of the AhR, in the last 20 years the scientific community has derived hypotheses about ligand binding from homology models based on known PAS-B domains in the bHLH-PAS protein family. Moreover, the lack of any structural information on the cytosolic complex (including Hsp90 and XAP2) recognized by the ligands has prevented the possibility of unveiling the AhR activation mechanism. Therefore, this work represents a milestone in the field and will contribute to great advancements in future mechanistic studies and drug design.

The Cryo-EM structure, given the good resolution, justify the conclusions advanced by the Authors about the Hsp90:AhR association mode. The proposed architecture, with the AhR PAS-A and PAS-B at the opposite sides of the Hsp90 middle-domain and a linker region interconnecting the two domains threaded through the Hsp90 dimer lumen, is also supported by the recently determined structures of Hsp90 in complex with other client proteins (Cdk4 and GR). Other Authors recently predicted this association mode, on the basis of both experimental evidences and the analogy with the Hsp90:Cdk4 complex, and, on these bases, they advanced an interesting hypothesis on the whole AhR activation and transformation mechanism (Soshilov et al. 2020, ref 22).

Moreover, the Hsp90-XAP2-AhR structure revealed for the first time the role of XAP2, that stabilizes the complex by binding both to Hsp90 and AhR.

Finally, the AhR PAS-B fold here determined confirms the characteristics of the structure predicted by many Authors on the basis of previous homology modelling studies. This has to be confirmed by more detailed analyses of the deposition in the PDB. Also the binding mode of indirubin, here detected, may represent a reference point for confirming/questioning the results of the plethora of computational molecular docking studies on different ligands based on AhR homology models. The microsecond-long molecular dynamics simulations performed by the Authors were useful both to refine the ligand binding mode and to estimate the relative energetic contributions of the key interacting amino-acids, that were confirmed by mutagenesis studies.

For what concerns my expertise, the methodology used is sound and meet the expected standards in this field. Moreover, there is enough detail provided in the methods for the work to be reproduced.

On the basis of the comments reported above, I think that the manuscript deserves publication.

Reviewer #2 (Remarks to the Author):

In general, a number of ligand-free AhRs in the cytoplasm have been reported to form a complex with HSP90-XAP2-p23 and to exist stably.

Ligand-bound AhR-chaperone complexes migrate to the nucleus, where AhR dissociates from the chaperone complex, forms a complex with Arnt, binds to XRE and acts as a transcription factor.

Gruszczuk et al. determined the structure of the indirubin-binding HSP90-XAP2-AhR complex using Cryo-EM. A number of AhR mutants were expressed and purified to elucidate the AhR ligand-binding mechanism. In the Exploring AHR ligand-binding mechanism section of the text, the authors state that p23 is not required for this ligand complex from in vitro experiments based on the data in Fig. 3c, but this is poorly supported. The components of the AhR complex in indirubin-treated cells need to be clarified in detail. For example, the complex should be determined by immunoprecipitation, cell staining, PLA and FRET methods.

If p23 is not included in the HSP90-XAP2-AhR complex even in indirubin-added cells, then an indirubin-specific activation mechanism for the AhR can be proposed in this paper. However, since p23 acts to inhibit the ATPase activity of HSP90, other ligand-binding AhR activation mechanisms, including TCDD, would be expected to exist independently of the activation mechanism by indirubin.

If indirubin, like other ligands, also binds to the HSP90-XAP2-p23-AhR complex, further structural analysis of this complex is required, which may differ from the HSP90-XAP2-AhR activation mechanism proposed by the authors.

Reviewer #3 (Remarks to the Author):

Gruszczyk et al. tackled a critical problem in the AHR field, the AHR's latent form bound to protein chaperones. Albeit the structure determined already contains a ligand, it still provides key insights into AHR latency. The research lab holds strong expertise in single-particle cryo-EM. The structure solution and structural analysis are solid in general. The studies guided by the cryo-EM structure, however, raise several major concerns.

1. A few minor comments for the cryo-EM study and structural analysis are provided below:

1a. It would be helpful to provide a partial close-up stereo view showing AHR PAS-B bound to ligand and the interface to Hsp90 and XAP2.

1b. It would be helpful to show Fig. 3f in stereo.

1c. Has the unliganded complex been investigated using Cryo-EM? The unliganded complex seems reasonable to try first, as it was believed that, upon ligand binding, AHR would dissociate from protein chaperones.

1d. Would the author expect the chaperone-bound PAS-B to have different conformations with or without ligand?

1e. According to the method description, ATP-MgCl₂ was used to stabilize the close form of the complex. It was believed that ATP hydrolysis would facilitate the reaction cycle of Hsp90. By any chance, "ATP" is a typo for "ATP[S]". If not, a quick clarification would be helpful.

1f. In Figure 3d, please provide the resolution shell of the map for the ligand.

2. The mutagenesis and co-IP studies, as presented in Fig. 2d, 2f, and 2h, raise several critical concerns. The authors concluded that the hydrophobic residues of AHR are not essential for Hsp90 binding, while the polar interactions are crucial. This conclusion is misleading due to the following reasons:

First, mutations to the polar residues seriously affect the AHR expression level, while those to the hydrophobic residues seem the opposite. Therefore, it is crucial to normalize the level of binding to the AHR expression level and show the average and standard error from three repeats at the bottom of the co-IP data.

Second, the mutation of a single hydrophobic residue to alanine among multiple hydrophobic contacts could be well tolerated. The authors might consider different mutation variants.

Other concerns about these studies are the lack of clarity in method description and data presentations. For example, throughout the entire manuscript, there is no description of what antibody/protein bait was used for fishing out the binding partner in the co-IP experiments. Instead, the information must be described in methods and figure legends and labeled on the figure panels. Furthermore, the locations of the MW standards need to show for all the western blot data.

3. AHR was known to interact with ligands only in the presence of protein chaperones. Thus, MD simulation of PAS-B without protein chaperones to identify the ligand entry is not quite right and could be misleading.

Minor comments:

- BSA is broadly known as bovine serum albumin. Using it for "buried surface area" is a bit odd.

Reviewer #4 (Remarks to the Author):

This wonderful paper significantly moves the Ah receptor field forward. The most striking aspect of this paper is not in its elegant and detailed structural analysis, but rather how it reconciles almost forty years of biochemistry. Put another way, the power of the report lies in the fact that there were no real surprises. Each aspect of the complex structure is supported by some past abstraction based on a biochemical or molecular result imparted onto a crude drawing of the AHR and XAP2 etc. depicted as blocks or circles. Being able to visualize this structure in such beautiful detail will be a watershed for the field.

A few minor considerations are provided:

1. At first use, XAP2 should include its alternate names (AIP, ARA9)
2. At first use, DRE should be accompanied by its alternate names (XRE, AHRE)
3. In early text, mention Sf9 expression was mode of production.
4. For clarity, "nucleotide" bound could be ATP bound?

A few thoughts:

I lost track of p23. Some discussion of why it doesn't appear in structure could be clearer.

The statement that Molybdate mechanism was unclear seems a bit overstated. The work of Toft and Pratt many years ago seem to point at the exact mechanism described here.

Stating why others failed at structural analysis seem like conjecture. What caught my eye is that A and B are separated by Hsp90 probably preventing aggregation.

Interesting that Sf9 worked when in the past for AHR and GR most protein was found in aggregates even with Hsp90 coexpression.

The “dynamic nature” of A and bHLH is in keeping with prior mapping of Hsp interactions.

Comment on use of 1-437. It is not clear that this fragment exhibits “normal” pharmacology for ligands.

Cryo-EM structure of the agonist-bound Hsp90-XAP2-AhR cytosolic complex
(Point-by-point response to the reviewers' comments)

Reviewer #1:

In the manuscript, the 2.85 Å Cryo-EM structure of the Hsp90-XAP2-AhR complex, bound to the indirubin ligand, is presented and analyzed.

A number of noteworthy information related to the structure and interactions of this complex are revealed in this work. These structural evidences have represented unanswered questions in the research related to the ligand-activated mechanism of the Aryl hydrocarbon Receptor (AhR) for many years. Given the key regulatory role that this transcription factor plays in modulating the ability of diverse chemicals to produce a wide spectrum of biological and toxicological effects, combined with its identification as a significant target for the development of therapeutic drugs, an understanding of its molecular mechanism is of great relevance.

As reported in extensive well-established literature, due to the lack of experimental structural information on the PAS-B ligand binding domain of the AhR, in the last 20 years the scientific community has derived hypotheses about ligand binding from homology models based on known PAS-B domains in the bHLH-PAS protein family. Moreover, the lack of any structural information on the cytosolic complex (including Hsp90 and XAP2) recognized by the ligands has prevented the possibility of unveiling the AhR activation mechanism. Therefore, this work represents a milestone in the field and will contribute to great advancements in future mechanistic studies and drug design.

The Cryo-EM structure, given the good resolution, justify the conclusions advanced by the Authors about the Hsp90:AhR association mode. The proposed architecture, with the AhR PAS-A and PAS-B at the opposite sides of the Hsp90 middle-domain and a linker region interconnecting the two domains threaded through the Hsp90 dimer lumen, is also supported by the recently determined structures of Hsp90 in complex with other client proteins (Cdk4 and GR). Other Authors recently predicted this association mode, on the basis of both experimental evidences and the analogy with the Hsp90:Cdk4 complex, and, on these bases, they advanced an interesting hypothesis on the whole AhR activation and transformation mechanism (Soshilov et al. 2020, ref 22).

Moreover, the Hsp90-XAP2-AhR structure revealed for the first time the role of XAP2, that stabilizes the complex by binding both to Hsp90 and AhR.

Finally, the AhR PAS-B fold here determined confirms the characteristics of the structure predicted by many Authors on the basis of previous homology modeling studies. This has to be confirmed by more detailed analyses of the deposition in the PDB. Also the binding mode of indirubin, here detected, may represent a reference point for confirming/questioning the results of the plethora of computational molecular docking studies on different ligands based on AhR homology models. The microsecond-long molecular dynamics simulations performed by the Authors were useful both to refine the ligand binding mode and to estimate the relative energetic contributions of the key interacting amino-acids, that were confirmed by mutagenesis studies.

For what concerns my expertise, the methodology used is sound and meet the expected standards in this field. Moreover, there is enough detail provided in the methods for the work to be reproduced.

On the basis of the comments reported above, I think that the manuscript deserves publication.

Response: We would like to thank the Reviewer for the positive comments on our work.

Reviewer #2:

In general, a number of ligand-free AhRs in the cytoplasm have been reported to form a complex with HSP90-XAP2-p23 and to exist stably.

Ligand-bound AhR-chaperone complexes migrate to the nucleus, where AhR dissociates from the chaperone complex, forms a complex with Arnt, binds to XRE and acts as a transcription factor.

Gruszczyk et al. determined the structure of the indirubin-binding HSP90-XAP2-AhR complex using Cryo-EM. A number of AhR mutants were expressed and purified to elucidate the AhR ligand-binding mechanism. In the Exploring AHR ligand-binding mechanism section of the text, the authors state that p23 is not required for this ligand complex from in vitro experiments based on the data in Fig. 3c, but this is poorly supported. The components of the AhR complex in indirubin-treated cells need to be clarified in detail. For example, the complex should be determined by immunoprecipitation, cell staining, PLA and FRET methods.

If p23 is not included in the HSP90-XAP2-AhR complex even in indirubin-added cells, then an indirubin-specific activation mechanism for the AhR can be proposed in this paper. However, since p23 acts to inhibit the ATPase activity of HSP90, other ligand-binding AhR activation mechanisms, including TCDD, would be expected to exist independently of the activation mechanism by indirubin.

If indirubin, like other ligands, also binds to the HSP90-XAP2-p23-AhR complex, further structural analysis of this complex is required, which may differ from the HSP90-XAP2-AhR activation mechanism proposed by the authors.

Response: We thank the Reviewer for seeking clarification on the importance of the p23 protein in the formation of AHR complex. Initially, we co-expressed all four proteins together, AHR, Hsp90, XAP2 and p23. While purifying the complex we quickly realized that there exist in fact two different complexes that can be easily separated from each other. The ternary complex Hsp90-XAP2-AHR devoid of p23 and a quaternary complex containing all four proteins Hsp90-XAP2-AHR-p23 (see Supplementary Fig. 1d and f). We compared the activity of both complexes using nano-DSF-based ligand binding assay. Both complexes displayed a thermal stabilization in the presence of the ligand indicating binding event.

Moreover, the initial characterisation of the quaternary Hsp90-XAP2-AHR-p23 complex using cryo-EM indicated a high heterogeneity of the sample due to the presence of several complexes with different number and location of p23 molecules. Therefore, in order to facilitate data analysis, we decided to use the ternary complex Hsp90-XAP2-AHR. The indirubin was therefore added to the Hsp90-XAP2-AHR sample followed by immediate preparation of the cryo-EM grids. It is worth noting that the entire system appears to be a very dynamic ensemble. Thus certain steps were necessary to make it suitable for structural analysis, like fixing Hsp90 in a closed conformation with molybdate and removing the highly dynamic p23 partner, for example.

In conclusion, our experiments do not suggest that the cytosolic complex to which indirubin binds is different from that involved in the mechanism of activation of other AHR agonists like for example TCDD. As explained above, p23 was purposely excluded from the final

complex for practical/technical reasons. A few sentences that explain why p23 is not present in the final structure have been added in the revised version (page 4).

We also agree with the Reviewer that the structural analysis of the Hsp90-XAP2-AHR-p23 complex would be very exciting. However, it is currently beyond the scope of our manuscript.

Reviewer #3:

Gruszczyk et al. tackled a critical problem in the AHR field, the AHR's latent form bound to protein chaperones. Albeit the structure determined already contains a ligand, it still provides key insights into AHR latency. The research lab holds strong expertise in single-particle cryo-EM. The structure solution and structural analysis are solid in general. The studies guided by the cryo-EM structure, however, raise several major concerns.

1. A few minor comments for the cryo-EM study and structural analysis are provided below:

1a. It would be helpful to provide a partial close-up stereo view showing AHR PAS-B bound to ligand and the interface to Hsp90 and XAP2.

Response: As requested by the Reviewer, we have included a stereo-view showing AHR PAS-B domain bound to the ligand and the interface to Hsp90 and XAP2 in the Supplementary Fig. 4c. Additionally, we also included a 360° view of the entire complex in Supplementary Movie 1 and of the PAS-B interface in Supplementary Movie 2.

1b. It would be helpful to show Fig. 3f in stereo.

Response: As requested by the Reviewer, we have included the stereo view of the ligand binding site in the Supplementary Fig. 8c. We also prepared a Supplementary Movie 3 that shows a 360° view of the ligand binding site.

1c. Has the unliganded complex been investigated using Cryo-EM? The unliganded complex seems reasonable to try first, as it was believed that, upon ligand binding, AHR would dissociate from protein chaperones.

Response: We thank the Reviewer for seeking clarification on the structural characterisation of the unliganded complex. Indeed, we collected and processed the data for the protein sample without addition of the ligand. The final map revealed the presence of an unidentified electron density in the binding site of AHR indicating a presence of a ligand that co-purified with the complex. We are currently trying to identify the unknown ligand and establish a biological relevance of this fact.

1d. Would the author expect the chaperone-bound PAS-B to have different conformations with or without ligand?

Response: We can speculate that the PAS-B domain in its ligand-bound state would adopt a somehow different conformation compared with the apo state. However, the exact extent to which these two structures would differ is very difficult to predict.

1e. According to the method description, ATP-MgCl₂ was used to stabilize the close form of the complex. It was believed that ATP hydrolysis would facilitate the reaction cycle of Hsp90. By any chance, "ATP" is a typo for "ATP_γS". If not, a quick clarification would be helpful.

Response: This point is discussed in detail in the section “Sodium molybdate mimics ATP γ -phosphate”, pages 7-8 of the revised manuscript. The AHR-Hsp90-XAP2 complex was prepared in the presence of ATP, MgCl₂ and sodium molybdate that is known to stabilize the closed form of the Hsp90 dimer and consequently its interaction with the client protein (AHR in this case). Our structural analysis reveals that the nucleotide binding site of each Hsp90 chain contains one ADP molecule (obtained from ATP hydrolysis), one magnesium ion, and one molybdate ion at the position normally occupied by the γ -phosphate of ATP. Molybdate resembles phosphate and is involved in contacts with Hsp90 residues similar to those of the ATP γ -phosphate. Thus we propose that by mimicking the ATP γ -phosphate upon ATP hydrolysis into ADP, the molybdate ion holds the Hsp90 dimer in its closed form induced by ATP.

1f. In Figure 3d, please provide the resolution shell of the map for the ligand.

Response: As requested by the Reviewer, we have provided directly in the Figure the resolution shell of the map for the ligand (0.0224).

2. The mutagenesis and co-IP studies, as presented in Fig. 2d, 2f, and 2h, raise several critical concerns. The authors concluded that the hydrophobic residues of AHR are not essential for Hsp90 binding, while the polar interactions are crucial. This conclusion is misleading due to the following reasons:

First, mutations to the polar residues seriously affect the AHR expression level, while those to the hydrophobic residues seem the opposite. Therefore, it is crucial to normalize the level of binding to the AHR expression level and show the average and standard error from three repeats at the bottom of the co-IP data.

Response: We thank the Reviewer for seeking clarification on the co-IP data. As suggested by the Reviewer we have performed quantification/normalization of our data which is reported in the Supplementary Fig. 6. The numerical analysis further highlights the relative weight of the various interfaces and the pivotal role of some residues compared to others in the complex formation. We also acknowledge the fact that all immunoprecipitation-based studies are not strictly quantitative and provide only a trend rather than absolute values to characterize protein-protein interactions and the role of individual residues in these processes. As the complex cannot be reconstituted from the individual components our choice of methods is limited.

According to the above remarks, the text describing co-IP experiments and the data obtained has been revised (pages 7, 8 and 9).

Second, the mutation of a single hydrophobic residue to alanine among multiple hydrophobic contacts could be well tolerated. The authors might consider different mutation variants.

Response: The goal of our mutational studies was to elucidate the importance of selected residues for the stability of the complex and not to completely disrupt its formation. We agree that mutation of a single amino acid residue might not have such a profound effect on the complex formation like mutation of several residues. And that is exactly what we observe, some residues seem to be redundant while others are playing an important role.

Other concerns about these studies are the lack of clarity in method description and data presentations. For example, throughout the entire manuscript, there is no description of what antibody/protein bait was used for fishing out the binding partner in the co-IP experiments. Instead, the information must be described in methods and figure legends and labeled on the figure panels. Furthermore, the locations of the MW standards need to show for all the western blot data.

Response: We thank the Reviewer for seeking clarification on the immunoprecipitation studies. In order to address this issue we added a short description of the method in the main text (page 7). The paragraph now reads:

To this end, we co-transfected HEK cells with plasmids expressing AHR fused with MBP- and Strep-tag, Hsp90 fused with FLAG-tag and XAP2 fused with MYC-tag. The complex was then pulled-down using magnetic beads conjugated with anti-MBP antibody. The presence of the particular components of the complex was detected using the corresponding antibodies.

The detailed description of the Co-IP experiment has been provided in the Materials & Methods section. We also provided the location of the molecular weight standards for all the Western blot membranes in the Source Data file.

3. AHR was known to interact with ligands only in the presence of protein chaperones. Thus, MD simulation of PAS-B without protein chaperones to identify the ligand entry is not quite right and could be misleading.

Response: As suggested by the Reviewer, MD simulations have been carried out using the whole complex instead of the isolated AHR PAS-B domain. The results obtained lead us to draw the same conclusions and are displayed in the Supplementary Fig. 10c-e.

We also provided an additional Supplementary Fig. 10a and b to further visualize the location of the entry site and highlight the fact that it is not obscured by the presence of the chaperone and co-chaperone proteins.

Minor comments:

- BSA is broadly known as bovine serum albumin. Using it for "buried surface area" is a bit odd.

Response: As requested by the Reviewer, we have removed the confusing abbreviation from the text and replaced it by the full term 'buried surface area'.

Reviewer #4:

This wonderful paper significantly moves the Ah receptor field forward. The most striking aspect of this paper is not in its elegant and detailed structural analysis, but rather how it reconciles almost forty years of biochemistry. Put another way, the power of the report lies in the fact that there were no real surprises. Each aspect of the complex structure is supported by some past abstraction based on a biochemical or molecular result imparted onto a crude drawing of the AHR and XAP2 etc. depicted as blocks or circles. Being able to visualize this structure in such beautiful detail will be a watershed for the field.

A few minor considerations are provided:

1. At first use, XAP2 should include its alternate names (AIP, ARA9)

Response: As requested by the Reviewer, we have included the alternative names of the XAP2 protein. The corresponding phrase now reads:

In the absence of ligand, AHR resides in the cytoplasm forming a complex with several other partners including heat shock protein 90 (Hsp90) and co-chaperones like X-associated protein 2 (XAP2, also known as AIP or ARA9) and p23.

2. At first use, DRE should be accompanied by its alternate names (XRE, AHRE)

Response: As requested by the Reviewer, we have provided the alternate names for DRE elements. The corresponding phrase now reads:

The newly formed heterodimer binds to the so-called “dioxin-response element” (DRE, also known as XRE or AHRE) DNA sequences and regulates the expression of target genes.

3. In early text, mention Sf9 expression was mode of production.

Response: As requested by the Reviewer, we have included a mention of the Sf9 expression system in the text. The corresponding phrase now reads:

Using the Sf9 insect cell expression system, we co-expressed a fragment of human AHR (residues 1-437) in the presence of Hsp90, XAP2 and p23, and purified the Hsp90-XAP2-AHR complex.

4. For clarity, “nucleotide” bound could be ATP bound?

Response: We use the generic term “nucleotide” rather than ATP because in our structure the ATP binding site is occupied by an ADP molecule and a molybdate ion mimicking the γ -phosphate of ATP.

A few thoughts:

I lost track of p23. Some discussion of why it doesn't appear in structure could be clearer.

Response: As now explained in the revised version of the manuscript, p23 was purposely excluded from the final complex for practical/technical reasons. A few sentences that explain why p23 is not present in the final structure have been added (page 4).

The statement that Molybdate mechanism was unclear seems a bit overstated. The work of Toft and Pratt many years ago seem to point at the exact mechanism described here.

Response: To the best of our knowledge, the fact that molybdate could “bind to an ATP site on Hsp90” was proposed as an hypothesis in a review article (Pratt & Toft, Endocrine Reviews, 1997) with no experimental validation. Our work not only brings experimental evidences that indeed molybdate binds to the ATP binding site, but it also provides additional mechanistic details revealing that molybdate can bind only to the ADP-bound form of Hsp90, at the position normally occupied by the γ -phosphate of ATP. By mimicking the presence of

the ATP molecule, the ADP/molybdate complex stabilizes the ATP-dependent closed conformation of the Hsp90 dimer.

Stating why others failed at structural analysis seem like conjecture. What caught my eye is that A and B are separated by Hsp90 probably preventing aggregation.

Response: We agree with the Reviewer that Hsp90 is stabilizing the AHR protein and preventing it from aggregation. However the reasons for the instability of AHR PAS-B compared to other protein PAS-B domains remain elusive, but most likely it is a result of different amino acid composition, especially hydrophobic residues present on the surface of the protein

Interesting that Sf9 worked when in the past for AHR and GR most protein was found in aggregates even with Hsp90 coexpression.

Response: Note however that in our case, AHR was co-expressed with chaperones (Hsp90) and co-chaperones (XAP2, p23) in Sf9 cells.

The “dynamic nature” of A and bHLH is in keeping with prior mapping of Hsp interactions.

Response: In the discussion, we suggest that the high dynamics of the N-terminal region harbouring the nuclear localisation signal (NLS) may have a functional relevance.

Comment on use of 1-437. It is not clear that this fragment exhibits “normal” pharmacology for ligands.

Response: We thank the Reviewer for seeking clarification on the used AHR sequence. In our construct we included the part of the protein corresponding to the structured part of the protein (N-terminus). We did not include the C-terminus as it is predicted to be mostly disordered and involved in the interaction with different co-activators. The choice of the sequence was partially guided by the sequence alignment with the structure of other members of the PAS protein family but the exact C-terminal boundary was a quite arbitrary decision. Our structural and thermal shift assays (nano-DSF) data confirm that our AHR fragment is fully competent to bind ligands.

REVIEWERS' COMMENTS

Reviewer #1 (Remarks to the Author):

I did not raised any concerns in my first review.

I read the revised paper and I think the Authors have answered the questions posed by the other reviewers.

Reviewer #2 (Remarks to the Author):

You have politely answered my questions about the AhR-HSP90-XAP-2-p23 complex.

In the near future, I look forward to your structural analysis of the full-length AhR and HSP90-XAP-2-p23 complexes.

Once the structural analysis has been carried out, we will be able to understand how the chaperone is involved in the ligand-binding properties and activation of the AhR, and how the chaperone is involved in the activation of the AhR.

Reviewer #3 (Remarks to the Author):

The authors made thorough responses to the comments. A minor note, the MW markers for western blots in Fig. 3 would be helpful. They are still missing in the revised manuscript.

Reviewer #4 (Remarks to the Author):

I accepted the ms on the first review. I had no required changes.

Cryo-EM structure of the agonist-bound Hsp90-XAP2-AHR cytosolic complex
(Point-by-point response to the reviewers' comments)

Reviewer #1:

I did not raise any concerns in my first review.

I read the revised paper and I think the Authors have answered the questions posed by the other reviewers.

We thank the reviewer for their positive and supportive comments.

Reviewer #2:

You have politely answered my questions about the AhR-HSP90-XAP-2-p23 complex. In the near future, I look forward to your structural analysis of the full-length AhR and HSP90-XAP-2-p23 complexes. Once the structural analysis has been carried out, we will be able to understand how the chaperone is involved in the ligand-binding properties and activation of the AhR, and how the chaperone is involved in the activation of the AhR.

We thank the reviewer for their constructive comments which have been very useful in revising this manuscript. The structural analysis of the quaternary AHR-Hsp90-XAP2-p23 complex is ongoing.

Reviewer #3:

The authors made thorough responses to the comments. A minor note, the MW markers for western blots in Fig. 3 would be helpful. They are still missing in the revised manuscript.

We thank the reviewer for their constructive comments which have been very useful in revising this manuscript. As suggested by the reviewer, we have added the molecular weight markers in the revised Fig. 3d, f and h.

Reviewer #4:

I accepted the ms on the first review. I had no required changes.

We thank the reviewer for their constructive comments which have been very useful in revising this manuscript.